# Determination of a Tumor-Promoting Microenvironment in Recurrent Medulloblastoma: A Multi-Omics Study of Cerebrospinal Fluid

**DOI:** 10.3390/cancers12061350

**Published:** 2020-05-26

**Authors:** Bernd Reichl, Laura Niederstaetter, Thomas Boegl, Benjamin Neuditschko, Andrea Bileck, Johannes Gojo, Wolfgang Buchberger, Andreas Peyrl, Christopher Gerner

**Affiliations:** 1Institute of Analytical Chemistry, Johannes Kepler University, Altenberger Strasse 69, 4040 Linz, Austria; bernd.reichl@jku.at (B.R.); Thomas.Boegl@jku.at (T.B.); 2Department of Analytical Chemistry, University of Vienna, Waehringer Straße 38, 1090 Vienna, Austria; laura.niederstaetter@univie.ac.at (L.N.); benjamin.neuditschko@univie.ac.at (B.N.); andrea.bileck@univie.ac.at (A.B.); 3Joint Metabolome Facility, Faculty of Chemistry, University of Vienna, Waehringer Straße 38, 1090 Vienna, Austria; 4Department of Pediatrics and Adolescent Medicine, Medical University of Vienna, Waehringer Guertel 18-20, 1090 Vienna, Austria; johannes.gojo@meduniwien.ac.at

**Keywords:** cerebrospinal fluid, oxylipins, hypoxia, lipidomics, mass spectrometry, medulloblastoma, metabolomics, multi-omics, polarized macrophages, proteomics

## Abstract

Molecular classification of medulloblastoma (MB) is well-established and reflects the cell origin and biological properties of tumor cells. However, limited data is available regarding the MB tumor microenvironment. Here, we present a mass spectrometry-based multi-omics pilot study of cerebrospinal fluid (CSF) from recurrent MB patients. A group of age-matched patients without a neoplastic disease was used as control cohort. Proteome profiling identified characteristic tumor markers, including FSTL5, ART3, and FMOD, and revealed a strong prevalence of anti-inflammatory and tumor-promoting proteins characteristic for alternatively polarized myeloid cells in MB samples. The up-regulation of ADAMTS1, GAP43 and GPR37 indicated hypoxic conditions in the CSF of MB patients. This notion was independently supported by metabolomics, demonstrating the up-regulation of tryptophan, methionine, serine and lysine, which have all been described to be induced upon hypoxia in CSF. While cyclooxygenase products were hardly detectable, the epoxygenase product and beta-oxidation promoting lipid hormone 12,13-DiHOME was found to be strongly up-regulated. Taken together, the data suggest a vicious cycle driven by autophagy, the formation of 12,13-DiHOME and increased beta-oxidation, thus promoting a metabolic shift supporting the formation of drug resistance and stem cell properties of MB cells. In conclusion, the different omics-techniques clearly synergized and mutually supported a novel model for a specific pathomechanism.

## 1. Introduction

Medulloblastoma (MB) is an embryonal tumor of the cerebellum, representing the most common malignant brain tumor in children [1]. MBs show a high tendency toward leptomeningeal dissemination, especially at recurrence, which occurs in up to 30% of children with standard risk MB [2,3]. Recurrent MB carries a very poor prognosis, with less than 10% survival, despite intensive treatment consisting of re-resection, high dose chemotherapy and re-irradiation [3]. In recent years, constant progress in molecular technologies has provided significant advancements in our understanding of MB. There is now consensus on four distinct molecular subgroups: wingless (WNT), sonic hedgehog (SHH), group 3, and group 4 [4]. Further refinement of these subgroups revealed numerous subtypes of MB, which already influenced current therapy concepts [5].

Despite the progress in molecular characterization, the biology and influence of the microenvironment in MB and MB recurrence is still poorly understood. The tumor microenvironment has become well recognized as a key factor in cancer progression, the promotion of metastasis, mediation of resistance against therapeutic drugs, and modulation of immune response [6,7,8]. The application of postgenomic methodologies, such as proteomics, metabolomics and lipidomics, may facilitate an integrated view on the phenotype of the tumor and its microenvironment. Proteomics, the systematic large-scale study of proteins, is regarded as one of the most potent tools in biomedical research. This analytical approach enables a comprehensive characterization of molecular mechanisms within an organism, including the identification of novel biomarkers for diagnostic and clinical uses [9]. Indeed, the proteome profiling of cerebrospinal fluid (CSF) has already identified relevant biomarkers for MB [10,11,12]. Downstream of proteomics, metabolomics has emerged as a complementary discipline, dealing with the global study of low molecular weight metabolites. The metabolism of the brain in different pathogeneses has been intensively investigated in recent years [13]. Studies in body fluids, like cerebrospinal fluid (CSF), plasma, urine and saliva suggest a significant disruption of the amino acid metabolism in MB, as well as in meningioma [13,14]. It has been shown that metabolites, especially of the tryptophan and methionine metabolism, are highly influenced by the development of tumors [13]. Among the whole range of metabolites, lipids stand out, due to their enormous diversity in structures and functionalities. Lipids play essential roles in cellular metabolism and have received growing attention in recent years, due to their correlation with several diseases, such as cardiovascular disease, inflammatory diseases, neurological disorders and cancer [15]. Long-chain and highly unsaturated phosphatidylcholine (PC) species, for example, have been reported to be significantly down-regulated in the serum of metastatic melanoma patients, suggesting the formation of platelet activating factors [16]. Similar observations were made in ovarian cancer, also showing a significant reduction of (poly-) unsaturated glycerophospholipids in patients with short survival time compared to healthy controls and patients with long survival time [17]. Oxylipins, a fatty acyl subclass, are key signaling molecules. They are derived from polyunsaturated fatty acids (PUFA), such as arachidonic acid (AA), comprise hundreds of individual bioactive compounds, and play important roles in inflammatory processes [18]. An overexpression of eicosanoid-producing enzymes, like cyclooxygenases (COX-1 and COX-2) and lipoxygenases (5-LOX, 12-LOX and 15-LOX) in gliomas and meningiomas, suggesting the promotion of tumorigenesis, has been reported before [19]. Moreover, increased levels of prostaglandin E2 (PGE2) have been detected in malignant brain tumors and suggested to play an important role in MB growth [20].

The underlying cellular and molecular mechanisms of MB and their leptomeningeal metastases are still poorly understood. The application of complementary omics approaches, often referred to as multi-omics strategies, provides an excellent opportunity to get an integrative understanding of the pathophysiology of recurrent MB and the intercellular crosstalk at the site of disease. In case of MB, and particularly regarding leptomeningeal dissemination, cerebrospinal fluid (CSF) represents a proximal fluid, defined as a biofluid that is located close or even in direct contact with the site of disease. In tumors, proximal fluids are enriched in compounds like proteins or peptides, which are secreted or released from adjacent tumor tissue, making these fluids a highly attractive source for biomarker discovery [21]. Most importantly, the CSF may uncover not only striking activities of tumor cells, but also of the microenvironment and the whole organism. The objective of this collaborative pilot study was thus to characterize protein, metabolite and lipid patterns in CSF from patients suffering from recurrent MB, in order to learn more about the intercellular crosstalk at the tumor site and to identify potential targets and biochemical pathways for further large-scale studies.

## 2. Materials and Methods

### 2.1. Subjects and Samples

CSF samples were obtained from 8 patients diagnosed with recurrent MB patients just before the onset of therapy at the Department of Pediatrics and Adolescent Medicine, Medical University of Vienna. CSFs were sampled during routine clinical procedures from Ommaya reservoirs. In addition, 7 CSF samples from age-matched patients without a neoplastic disease were analyzed for reference purposes. CSF samples were snap-frozen without further processing and stored at −80 °C until sample preparation. The study was approved by the institutional review board of the Medical University of Vienna (EK 1244/2016).

### 2.2. Sample Preparation and Instrumental Analysis

Lipids, proteins and metabolites were analyzed using distinct analytical workflows based on mass spectrometric analyses. These workflows were based on already published protocols [16,22,23,24] using MaxQuant [25] in case of proteomics and further optimized for each class of molecule. Therefore, even two complementary workflows were applied for lipid analysis, one dedicated for glycerophospholipids and sphingolipids, and the other for fatty acyls, referred to as “lipids” and “oxylipins”, respectively. A detailed description regarding sample preparation and instrumental analysis is given separately for each class of molecule in the Appendix A including Appendix A. The mass spectrometry proteomics data have been deposited to the ProteomeXchange Consortium (http://proteomecentral.proteomexchange.org) via the PRIDE partner repository [26] with the dataset identifier PXD018226 and 10.6019/PXD018226.

### 2.3. Lipid Terminology and Identification

The terminology of the analyzed lipids is based on the lipid species level terminology, as described before [27,28]. The applied identification level defines lipid category and class, as well as the sum of components in the attached fatty acyl chains and, in the case of sphingolipids, the sphingoid base (number of carbons and double bonds, e.g., PC 34:1; Appendix A). In the presence of more than one structural isomer differing in retention times, species are denoted as e.g., PC 36:2a and PC 36:2b. In the case of oxylipins, molecules were either identified referring to purchased standards or designated, according to nominal molecular mass and chromatographic retention time.

### 2.4. Data Analysis

The four data matrices obtained as described above were loaded into Perseus software (version 1.6.7.0) [29], followed by filtering for those analytes that were present in at least 70% of samples in at least one group (MB or Ref). Next, data were log 2 transformed, and missing values were replaced by normally distributed random numbers, with a set width of 0.3 and a downshift of 1.8. These data were visualized with a volcano plot for each of the four data matrices, in order to identify the significantly regulated molecules. A two-sided t-test was applied for statistical significance testing with the number of randomizations set to 250, FDR threshold set to 0.05 and S0 to 0.1. In addition, data were visualized with heatmaps and principal component analysis (PCA).

## 3. Results

The clinical diagnosis and relevant analytical data obtained from the routine laboratory test program of eight patients diagnosed with recurrent medulloblastoma (here designated as MB_1–8) and seven patients without neoplastic diseases serving as reference (Ref_1–7), are listed in Appendix A. All 15 patients were juvenile, with a mean age (± standard deviation) of 6.9 ± 4.0 years in MB and 9.7 ± 5.7 years in Ref patients at the time of sampling. All samples appeared optically clear, except for MB_6 (xanthochromic) and Ref_7 (turbid), resulting in their exclusion from statistical analyses in the case of lipids, oxylipins and amino acids. These two samples were also outstanding in terms of cell count (22 and 28 cells per µL, respectively) and protein concentration (301.1 and 96 mg dL^−1^, respectively). Apart from these two samples, samples MB_2 and Ref_4 were suspicious, with MB_2 showing a cell count of 8 cells per µL and a protein concentration of 218.3 mg dL^−1^, and Ref_4 a cell count of 37 cells per µL, but inconspicuous protein concentration of 28.8 mg dL^−1^. In the following, comparative molecular analyses regarding CSF samples from medulloblastoma patients and non-neoplastic references, respectively, will be presented. Heatmaps visualizing the abundance distributions of relevant molecules, principal component analyses (PCA) visualizing group separation efficiency and volcano plots highlighting molecules significantly deregulated in MB samples will be presented for each molecular class analyzed.

### 3.1. Lipidomics Results Indicate Increased Lipolysis

Overall, quantitative lipid levels showed high inter-individual variances (Figure 1A). Applying the restrictive conditions outlined in the Materials and Methods section for data analysis, 38 out of 59 investigated molecules were included in the statistical analysis (Appendix A). In addition to MB_6 and Ref_7, the sample MB_7 displayed highly increased lipid levels most probably derived from cell debris, as demonstrated in Figure 1A. Since it was attributed with clear sample appearance and insuspicious cell count and total protein concentration (Appendix A), the sample was included in the statistical analysis. Indeed, PCA successfully separated the neoplastic MB patients from the non-neoplastic reference group (Figure 1C), while only PE 34:1 showed significant regulation (Figure 1B). The efficient group separation seems to result from a common trend of most PC and PE species to be down-regulated in MB samples (Figure 1A,C), pointing to increased lipolysis in MB patients.

### 3.2. Analysis of Fatty Acids and Oxylipins Indicate a Prevalence for Lipoxygenase and Epoxygenase, but not Cyclooxygenase Products

As expected, oxylipins also showed substantial inter-individual variations and highlighted MB_6 and Ref_7 as outliers, resulting in their exclusion from further statistical analysis (Figure 2A). Including precursor PUFAs, a total of 22 molecules were identified reproducibly and thus quantitatively assessed (Appendix A). Linoleic acid (LA), otherwise readily detectable in serum samples, was undetectable. Three LOX-products were slightly up-regulated, whereas COX-products were hardly detectable (Appendix A). Again, PCA separated the two groups quite well (Figure 2C), and only a few molecules were found to be significantly regulated (Figure 2B). The molecule 12,13-DiHOME, an anti-inflammatory epoxygenase product [30], and a molecule isobaric to docosapentaenoic acid, were found to be significantly increased in MB patients (Figure 2B).

### 3.3. Proteome Profiling Deciphers an Anti-Inflammatory and Tumor-Promoting Microenvironment Potentially Originating from Immune Cells and Hypoxic Conditions

The largest number of molecules assessed in CSF samples was represented by a total of 729 proteins (Appendix A). The observed differences in protein abundances between the two groups were profound, as demonstrated by the heatmap (Figure 3A). PCA readily separated the groups (Figure 3C), and a total of 178 proteins was found to be significantly regulated (Figure 3B, Appendix A). Among these, known biomarkers for MB (FSTL5 [31]), and other tumor types such as neuroendocrine tumors (ENO2 [32], FMOD [33], ART3 [34], COL6A3 [35]) and PIP [36], were found to be significantly up-regulated (Figure 3B, dark green). A total of 55 regulated proteins were found to be associated with the gene ontology term “leukocyte mediated immunity [2443]” and/or “defense response [6952]”, (Figure 3B, brown), most of them characteristic for myeloid cells such as macrophages and strongly indicating an active involvement of immune cells. None of the up-regulated proteins were found to be related to inflammatory stimulated cells, as characterized by us previously [22,37]. In contrast, the detectable potential pro-inflammatory proteins MX2, SPP1 and PIBF1 were found to be down-regulated (Figure 3B, red), whereas the rather anti-inflammatory proteins ANXA1, LTBP1, SERPINA4 and TXN were found to be up-regulated in MB samples (Figure 3B, orange). Remarkable was the detection of a distinct molecular signature indicative for hypoxia, represented by the up-regulated proteins ADAMTS1 [38], GAP43 [39] and GPR37 [40] (Figure 3B, blue). The detailed classification of proteins is documented in Appendix A.

### 3.4. A Characteristic Metabolite Signature Indicates Hypoxic Conditions in CSF of MB Patients

Similar to the proteome profiling data, the metabolome data demonstrated effective group separation via heatmap (Figure 4A) as well as PCA (Figure 4C), pointing to a metabolic shift characteristic for the MB disease state. The amino acids tryptophan, tyrosine, methionine, lysine and serine were found to be significantly up-regulated, while proline, leucine and isoleucine were down-regulated (Figure 4B). Four of these amino acids, i.e., tryptophan, methionine, serine and lysine, were described to be induced in CSF upon hypoxia [41] and may thus indicate such conditions in CSF of MB patients. Remarkably, the tryptophan oxidation product characteristic for inflammatory activation, kynurenine [42], was positively identified, but not found to be regulated (Figure 4B), again indicating the absence of pro-inflammatory conditions.

## 4. Discussion

To the best of our knowledge, here we present the first multi-omics study of cerebrospinal fluid of patients with recurrent medulloblastoma, aiming at a better characterization of the pathophysiology of this disease. It appears justified to expect characteristic alterations of metabolites, lipids and proteins associated with such a severe pathophysiological state. This is a pilot study collecting detailed molecular data of a limited number of patients and shall answer the question of whether complex data derived from a mass spectrometry-based multi-omics study could support a better understanding of the pathophysiology of recurrent medulloblastoma.

Proteomics and metabolomics have the power of providing a completely unbiased view on the state of body fluids, such as CSF, associated with a specific disease. To date, research on molecular biology of medulloblastoma has mainly focused on elucidating the cell of origin and the associated genotype and DNA methylation pattern [43]. However, rather few data are available regarding the involvement of the tumor microenvironment, known to have the potential to strongly influence the course of disease [8]. This can be readily accomplished using the mass-spectrometry-based post-genomic techniques presently employed [44]. Indeed, most molecules detected in CSF may not be derived directly from the tumor cells, but also from the choroid plexus, neurons and other brain cells. Comparisons between non-neoplastic and MB patient samples are thus providing valuable information regarding the functional states of cells making up the tumor microenvironment.

The most striking conclusion supported by the present data was a predominance of anti-inflammatory molecules in CSF of MB patients. All detected proteins with potential pro-inflammatory activities were found to be down-regulated in MB samples, whereas all detected proteins with potential anti-inflammatory activities were found to be up-regulated (Figure 3B, Appendix A). Regarding metabolomics analyses, kynurenine, a product of indolamine 2,3-dioxygenase characteristic for inflammation [45], was found at low levels and not induced in MB samples. In addition, lipoxins derived from cyclooxygenase activity, known to result from inflammatory stimulation [46], were hardly detectable in all CSF samples. Thus, proteomics, lipidomics and metabolomics data proved to be independently consistent. At first glance, these findings seem to contradict the reported role of PGE2, a COX-product, for MB pathogenesis [20]. However, this report refers to MB cells directly, whereas here, we refer to CSF mainly made up by the tumor microenvironment. Actually, we interpret these data as an effort of the microenvironment to suppress inflammatory signals originating from the tumor cells. 

Another observation was a molecular signature indicative for hypoxia detected by proteome profiling (Figure 3B). In addition to the three significantly regulated proteins, the hypoxia-induced proteins CALB1 [47], as well as DDAH1 [48], were also found strongly up-regulated, yet not as uniform as the other three, thus lacking significance. This indication for MB-associated hypoxia was independently supported by the present metabolomics data demonstrating an up-regulation of tryptophan, methionine, serine and lysine (Figure 4B), in accordance with existing literature reporting a hypoxic adaptation of the cerebellum in association with the up-regulation of these amino acids [41]. This adaptive metabolic response most likely resulted from autophagy and mTOR signaling consequent to hypoxia [49,50]. The most strongly and significantly induced metabolite, tryptophan, has been reported to be abnormally high in cachexia [51], a syndrome frequently occurring in medulloblastoma [52]. We have observed that adaptation to hypoxic stress may result in tumor progression in the case of multiple myeloma [53] and ovarian cancer [54], compatible with rather tumor-promoting consequences of chronic hypoxic stress [55]. Taken together, proteomics and metabolomics data, independently of each other, support the hypothesis that hypoxia may also be characteristic for MB.

Thus, the question arises as to which cells might be responsible for establishing the anti-inflammatory conditions presently observed. Actually, a rather anti-inflammatory state is already characteristic for normal brain [56], involving TGF-beta and alternatively polarized macrophages [57]. Remarkably, tumor-associated macrophages derived from circulating monocytes or microglia are known to produce even more TGF-beta, further promoting the alternative polarization of macrophages and acting anti-inflammatory, thus further promoting tumor growth in various forms of brain tumors [58]. All four anti-inflammatory proteins (Figure 3B) were described by us previously to be secreted by peripheral leukocytes [22]. According to the Expression Atlas [59], the hypoxia-associated molecules CALB1, DDAH1, GAP43 and GPR37, as well as the tumor markers ART3, ENOL2 and FMOD and the tumor promoter NTNG1, are typically expressed in hematopoietic stem cells. It remains to be determined whether the specific MB microenvironment is shaped by myeloid precursor cells, which may subsequently give rise to tumor-associated macrophages, or whether MB cells eventually express such stem cell marker proteins. The tumor promoters BASP1, GPC1, PSAT1 and SH3BGRL are reported to be typically expressed in macrophages by the Expression Atlas. The most strongly up-regulated proteins in MB samples, such as LTF, DCD, CHIT1 and many more (Appendix A), are also specific for macrophages, clearly documenting a dominant contribution of myeloid cells to the MB microenvironment.

Fatty acid oxidation has been described to be characteristic for cancer stem cells [60] and essential for glioma and glioblastoma cell growth and proliferation [61,62], while hardly any experimental data are yet available regarding MB. In addition, macrophage polarization and anti-inflammatory activities are associated with increased fatty acid oxidation [63]. The resulting increased demand for fatty acids in MB is quite compatible with the present observation of an almost generally decreased level of lipids in CSF. In addition, macrophage polarization has been demonstrated to induce various P450 epoxygenases [64]. The oxylipin presently found to be most strongly up-regulated in MB was 12,13-DiHOME (Appendix A). This lipid hormone is generated from linoleic acid by the combined action of epoxygenases and epoxide hydrolases and has been described to mediate resolution of inflammation [30]. Linoleic acid and isobaric molecules were found decreased or undetectable in the CSF samples (Appendix A), compatible with rapid catabolism. Furthermore, 12,13-DiHOME is a PPAR-γ ligand strongly promoting beta oxidation [65,66], thus potentially representing a powerful driver of the above described metabolic adaptation detectable in CSF of MB patients.

Taken together, the present multi-omics data suggested a tumor promoting vicious cycle established by the MB-associated microenvironment (Figure 5). Hypoxic conditions may cause tumor cells to become more aggressive [53] and additionally induce the formation of polarized macrophages [67]. Furthermore, hypoxia has been described as key promotor of cancer stem cell resistance properties [68], and appears to be a central factor contributing to the aggressive biological behavior of MB. Polarized macrophages are capable of forming the lipid hormone 12,13-DiHOME, which was found to be strongly up-regulated in CSF of MB patients. A switch to cellular energy metabolism via beta-oxidation is strongly promoted by this lipid hormone and evidenced by the present lipidomics data. Increased beta-oxidation is a hallmark of cancer stem cells and polarized macrophages. The numerous anti-inflammatory proteins, as well as tumor promoters presently identified in the CSF of MB patients, identify alternatively polarized macrophages as predominant modulators of the MB microenvironment. Importantly, the present proteomics, as well as metabolomics data, strongly indicate hypoxic conditions in CSF of MB patients, which may also be driven by increased oxygen consumption due to beta oxidation. Actually, rescue from hypoxia has been described to require the expression of olig2, essential for functional myelination [69]. It is thus plausible to assume that the characteristic expression of olig2 by stem cell like progenitors giving rise to MB [70] is related to the consequences of this vicious cycle.

## 5. Conclusions

The present data derived from four different mass spectrometry-based multi-omics data depict a consistent and congruent pathophysiological state of the MB-associated microenvironment, compatible with the most current single cell transcriptomics data [70]. It may be justified to note that the presently observed molecular patterns were detectable, irrespective of the molecular subtype of MB. This may indicate characteristic functional alterations of the tumor microenvironment irrespective of the genetic subtype of the tumor cells. Future studies will be required to understand the course of events resulting in the establishment of the tumor-promoting microenvironment in MB patients, while the disruption of the described vicious cycle may represent an attractive therapeutic target.

## Figures and Tables

**Figure 1 cancers-12-01350-f001:**
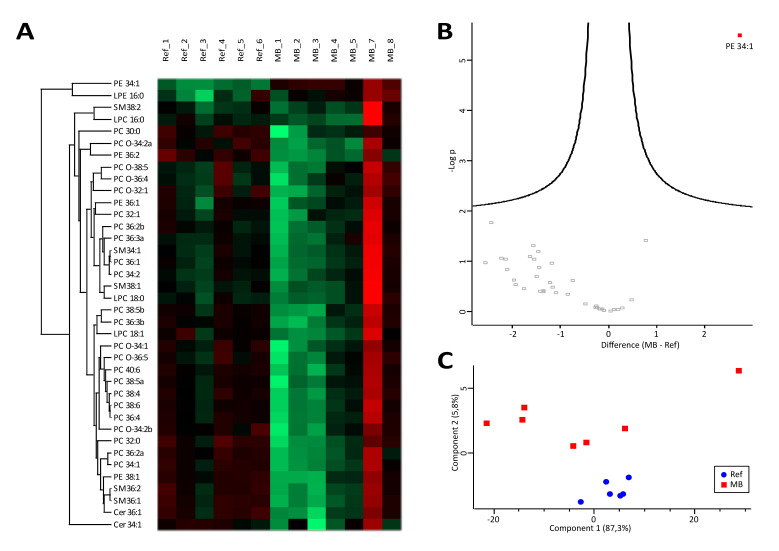
Lipids. Heatmap (**A**) and volcano plot (**B**) visualizing abundance distributions and significant differences of lipids, respectively, in cerebrospinal fluid (CSF) of medulloblastoma (MB) and reference (Ref) samples. Difference values in the volcano plot are shown in a logarithmic scale to the basis of 2. Principal component analysis (**C**) demonstrates group separation of neoplastic medulloblastoma (MB) and non-neoplastic reference (Ref) samples, based on investigated lipids.

**Figure 2 cancers-12-01350-f002:**
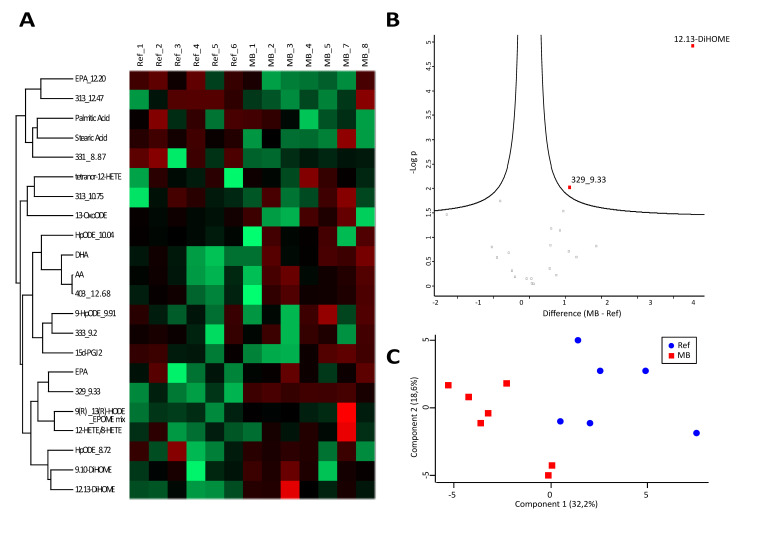
Oxylipins. Heatmap (**A**) and volcano plot (**B**) visualizing abundance distributions and significant differences of oxylipins, respectively, in cerebrospinal fluid (CSF) of medulloblastoma (MB) and reference (Ref) samples. Difference values in the volcano plot are shown in a logarithmic scale on the basis of 2. Principal component analysis (**C**) demonstrates group separation of neoplastic medulloblastoma (MB) and non-neoplastic reference (Ref) samples, based on investigated oxylipins.

**Figure 3 cancers-12-01350-f003:**
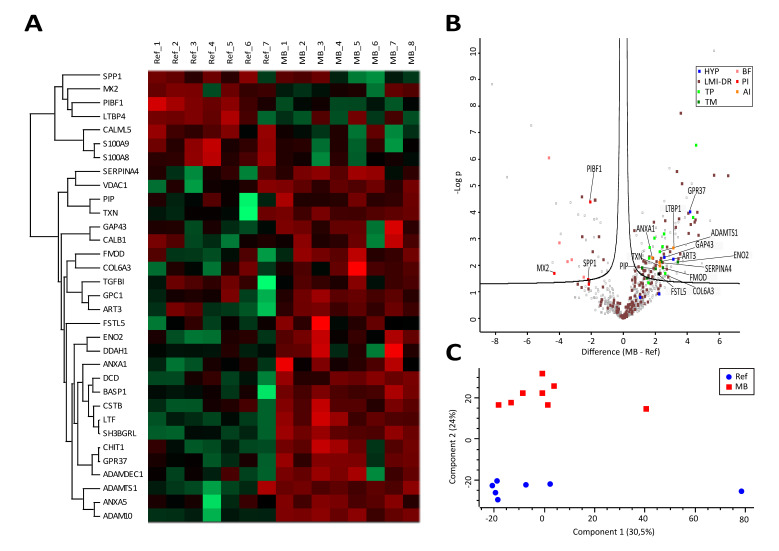
Proteins. Heatmap (**A**) and volcano plot (**B**), visualizing abundance distributions and significant differences of proteins, respectively, in cerebrospinal fluid (CSF) of medulloblastoma (MB) and reference (Ref) samples. Difference values in the volcano plot are shown in a logarithmic scale to the basis of 2. Proteins were additionally classified according to the gene ontology term “leukocyte mediated immunity [2443]” and/or “defense response [6952]” (LMI-DR), and according to existing literature into anti-inflammatory proteins (AI), proteins related to brain function (BF), hypoxia-related proteins (HYP), pro-inflammatory proteins (PI), tumor markers (TM) and tumor promotors (TP). Principal component analysis (**C**) demonstrates the group separation of neoplastic medulloblastoma (MB) and non-neoplastic reference (Ref) samples based on investigated proteins.

**Figure 4 cancers-12-01350-f004:**
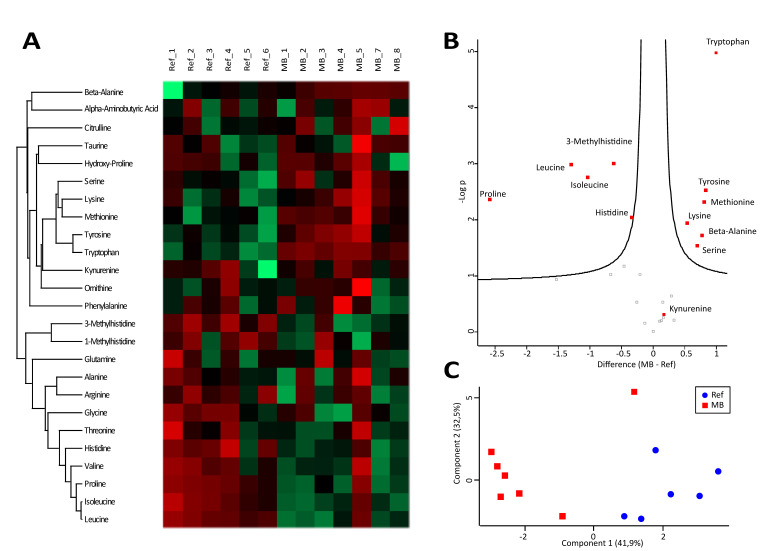
Amino Acids. Heatmap (**A**) and volcano plot (**B**) visualizing abundance distributions and significant differences of amino acids, respectively, in cerebrospinal fluid (CSF) of medulloblastoma (MB) and reference (Ref) samples. Difference values in the volcano plot are shown in a logarithmic scale, to the basis of 2. Principal component analysis (**C**) demonstrates group separation of neoplastic medulloblastoma (MB) and non-neoplastic reference (Ref) samples, based on investigated amino acids.

**Figure 5 cancers-12-01350-f005:**
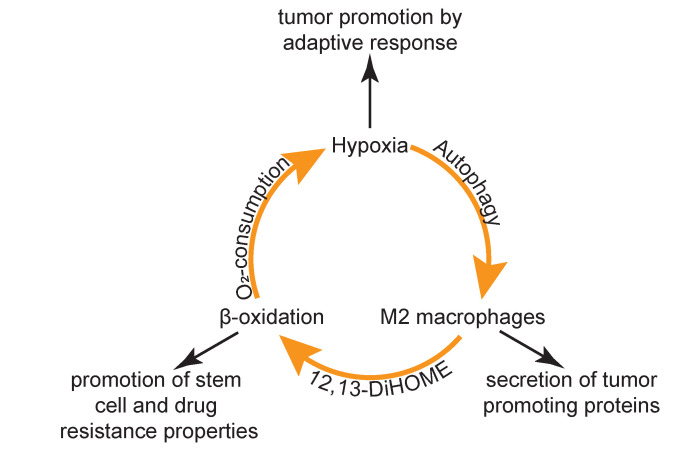
Model of tumor promoting vicious cycle. Hypoxia induces the formation of M2 macrophages via autophagy. M2 macrophages promote beta oxidation via 12,13-DiHOME. As beta oxidation requires increased oxygen consumption, it may aggravate hypoxia. Each of these entities contributes to tumor progression independently.

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
