# Peer review of "Determination of a Tumor-Promoting Microenvironment in Recurrent Medulloblastoma: A Multi-Omics Study of Cerebrospinal Fluid"

_cancers, 2020, doi:10.3390/cancers12061350_

Round 1

Reviewer 1 Report

Reichl 
B. et al. in their manuscript “Determination of a tumor-promoting 
microenvironment in recurrent medulloblastoma: a 
multi-omics study of cerebrospinal fluid”, 
characterized the protein, metabolite and lipid patterns in CSF from patients with recurrent MB. The manuscript is of interest in this research field because, as sustained by the authors, the application of multi-omics strategies represents an opportunity to get a deep understanding of the pathophysiology of this malignancy, focusing also on tumor microenvironment. Nevertheless, the work needs to be improved: the major criticism is related to the small number of samples analyzed. MB is a very heterogeneous tumor classified in 4 molecular subgroups differing from each other for many aspects. The samples number analyzed is not representative of each subgroup: 5 G4, 1 G3, 1SHH, 1 n.a., and this issue could is the reason of the high inter-individual variances observed by the authors in Figure 1A. This reviewer understands the difficulty to find patients MB samples, but the analysis of a larger number of samples could improve the relevance of this very interesting work.

Previous proteomics analysis of CSF from MB patients have revealed low levels of prostaglandin D2 synthase (Rajagopal M.U. et al. 2011), absence of hemoglobin subunit beta fragment (Desiderio C. et al. 2012) and high levels of polysialic-neural cell adhesion molecule (Figarella-Branger D. et al. 1996). Have the authors found the levels of these proteins modulated? The authors should discuss this aspect.

In order to improve heat maps, the authors should insert in the figures the hierarchical clustering of samples and the color scale bars.

Author Response

We want to thank reviewer 1 for positive and helpful comments. Please find enclosed our detailed answers.

(...)

The samples number analyzed is not representative of each subgroup: 5 G4, 1 G3, 1SHH, 1 n.a., and this issue could is the reason of the high inter-individual variances observed by the authors in Figure 1A.

  • We were not analysing tumor cells, but CSF which is rather representative for the tumor microenvironment. We think it is highly relevant to record the high level of uniformity of molecular alterations irrespective of the tumor classification. The high inter-individual variances of lipids can be observed in other body fluids as well and are most plausibly caused by some variations of lipoprotein particles and other particulate substances in CSF

This reviewer understands the difficulty to find patients MB samples, but the analysis of a larger number of samples could improve the relevance of this very interesting work.

  • we highly acknowledge this positive comment. However, we are now limited to the number of samples analyzed. We think the sample size is adequate for a pilot study as indicated.

Previous proteomics analysis of CSF from MB patients have revealed low levels of prostaglandin D2 synthase (Rajagopal M.U. et al. 2011), absence of hemoglobin subunit beta fragment (Desiderio C. et al. 2012) and high levels of polysialic-neural cell adhesion molecule (Figarella-Branger D. et al. 1996). Have the authors found the levels of these proteins modulated? The authors should discuss this aspect.

  • Thank you also for this very competent comment. Actually we have listed prostaglandin D2 synthase (P41222) in Suppl. Table 6, indicating there was no relevant difference (ln2diff=0.11). Hemoglobin subunit beta was clearly positively identified with 10 proteotypic peptides. The high levels of various forms of neural cell adehsion molecules are as well clearly reported in TabS6, all of them up-regulated in MB, but never coming close to significance.

In order to improve heat maps, the authors should insert in the figures the hierarchical clustering of samples and the color scale bars.

  • Actually our first version was exactly according to your suggestion. However, these clusters are rather poor in information and when working ourselves with the data we found it easier to compare one given patient across all data when using the chosen representation.

Reviewer 2 Report

The manuscript by Reichl et al. investigates the microenvironment of recurrent medulloblastoma. For that, they used a mass spectrometry-based multi-omics study of CSF from recurrent medulloblastoma patients. They characterized the proteome, lipidome and metabolome of the CSF. This study brings new molecular information of this pathology and new potential biomarkers. However, I think that more validation is needed to support the results.

Below are my comments:

  • Why analyzing the recurrent medulloblastoma only? Is something known about the microenvironment of the primary tumors? Authors should at least discuss this in the introduction.
  • How can you assign the soluble factors identified in the CSF to the tumor cells or to the microenvironment cell components? Tumor cells can also release immune related molecules and hypoxic molecules to modulate the microenvironment.
  • In the materials and methods section, in 2.2 you wrote “these workflows were based on already published protocols using MaxQuant in case of proteomics and further optimized for each class of molecule”. You used this software for protein analysis and not for the other molecules? It is a bit confusing.
  • Results section:
    • You found cells in the CSF of patients. What is the workflow for CSF preparation? Have you centrifuged the CSF before performing molecules purification in order to remove the cells? If not, you can have contamination from the circulation cells. I guess not since in paragraph 3.1 you said that “MB_7 displayed highly increased lipid levels most probably derived from cell debris”.
    • You said that MB_2 and Ref_4 were suspicious due to high protein concentration or high cell count. Did you remove these two samples from the analysis?
    • In the lipidomic analysis, you only found one lipid (PE 34:1) which is significantly regulated between control and MB groups. What is this lipid? You found a lot of lipids downregulated in MB samples, but is it significant?
    • The figures are of very low resolution, it is very difficult to read. Be careful.
    • For proteome analysis, you only present the differentially regulated proteins between the two groups. Have you looked at the unique proteins identified in each group? It would be interested to do.
    • You identified 55 regulated proteins associated to leukocyte mediated immunity. Have you listed them in a table somewhere? How did you perform this gene ontology analysis? It is not described in the materials and methods section.
    • In the CSF, you can find extracellular vesicles containing a lot of lipids, proteins, metabolites. The molecules you identified in your study can come from these elements. These vesicles are essential in cell-cell communication. Have you looked at the presence of these vesicles in the CSF?
    • The findings are interesting but need to be validated. For example, I suggest the authors to validate at least in the CSF the presence of immune molecules by ELISA assay. And compare the control vs the MB samples for the presence of pro-inflammatory vs anti-inflammatory molecules. Moreover, it would be great to perform some immunohistochemistry experiments on MB samples (FFPE tissues) to validate the presence of immune cells or hypoxic regions in the microenvironment of the tumor.

Author Response

We would like to thank reviewer 2 for helpful comments. Please find enclosed our point to point replies:

•    Why analyzing the recurrent medulloblastoma only? Is something known about the microenvironment of the primary tumors? Authors should at least discuss this in the introduction

  • Recurrent medulloblastoma shows the typical drug resistance features typical for this disease. We have considered this fact of sufficient relevance to make this well-defined choice

•    How can you assign the soluble factors identified in the CSF to the tumor cells or to the microenvironment cell components? Tumor cells can also release immune related molecules and hypoxic molecules to modulate the microenvironment.

  • Actually we have clearly pointed out, that a gene set enrichment analysis would suggest macrophages as main choice but does not proof so. We have use carefully chosen words in the discussion to make clear that also tumor cells might secrete such factors. Anyhow, also if it was the tumor cells the adaptation to the metabolic conditions would be evident. In addition, the finally cited transcriptomics paper also suggests it was mainly macrophages rather than tumor cells.

•    In the materials and methods section, in 2.2 you wrote “these workflows were based on already published protocols using MaxQuant in case of proteomics and further optimized for each class of molecule”. You used this software for protein analysis and not for the other molecules? It is a bit confusing.

  • We apologize if this was confusing. We stated we were using MaxQuant in case of proteomics and described all other workflows as well in detail. For conform visualisation we used Perseus for all kinds of data as indicated.

•    Results section:
o    You found cells in the CSF of patients. What is the workflow for CSF preparation? Have you centrifuged the CSF before performing molecules purification in order to remove the cells? If not, you can have contamination from the circulation cells. I guess not since in paragraph 3.1 you said that “MB_7 displayed highly increased lipid levels most probably derived from cell debris”.

  • No the CSF was not centrifuged. We hardly detected cytoplasmic proteins, thus ruling out relevant contamination with cells. We meant it was any kind of debris, not necessarily cell debris.

o    You said that MB_2 and Ref_4 were suspicious due to high protein concentration or high cell count. Did you remove these two samples from the analysis?

  • No, we did the analysis as indicated and only excluded optically unclear samples for lipidomics and metabolomics. While the lipid content strongly varied as visible in the corresponding heat-map 1A and 2A, the overall protein content did not differ in the less clear samples (as also visible in the heatmap Figure 3A).

o    In the lipidomic analysis, you only found one lipid (PE 34:1) which is significantly regulated between control and MB groups. What is this lipid? You found a lot of lipids downregulated in MB samples, but is it significant?

  • We think we have already described the necessary details. PE is listed in the abbreviations (Phosphatidylethanolamines), 34:1 refers to the number of carbons and double bonds of the sum of two fatty acids present in the molecule. We are afraid that no single down-regulation reached significance threshold (as depicted in Figure 1B and described in the text). Hwever, as obvious from the same Figure, the general statement that lipid levels are decreased in MB patients is justified.

o    The figures are of very low resolution, it is very difficult to read. Be careful.

  • The low resolution is only a property of the files for review. Obviously we have high-resolution images and only high-resolution images will be published.

o    For proteome analysis, you only present the differentially regulated proteins between the two groups. Have you looked at the unique proteins identified in each group? It would be interested to do.

  • This information is indeed important and available. However, the files containing all this information would be hard to handle. We have uploaded all data to ProteomeXchange, including raw data and analysis results and making it public available. So this and many other research questions can be answered referring to these data.

o    You identified 55 regulated proteins associated to leukocyte mediated immunity. Have you listed them in a table somewhere? How did you perform this gene ontology analysis? It is not described in the materials and methods section.

  • Yes. You can find the term "classification" in Table S5, all assignments are clearly documented

o    In the CSF, you can find extracellular vesicles containing a lot of lipids, proteins, metabolites. The molecules you identified in your study can come from these elements. These vesicles are essential in cell-cell communication. Have you looked at the presence of these vesicles in the CSF?

  • I fully agree with this notion. Our protocol results in the analysis of both soluble and insoluble and particulate matter, as it gets solubilized. Thus, the content of such particles is represented, but not distinguished as such.

o    The findings are interesting but need to be validated. For example, I suggest the authors to validate at least in the CSF the presence of immune molecules by ELISA assay. And compare the control vs the MB samples for the presence of pro-inflammatory vs anti-inflammatory molecules. Moreover, it would be great to perform some immunohistochemistry experiments on MB samples (FFPE tissues) to validate the presence of immune cells or hypoxic regions in the microenvironment of the tumor.

  • We do not agree with this notion. We have used high-resolution mass spectrometry using highly stringent conditions. It is clearly recognized that the reliability of hig-quality MS results by far exceeds the reliability of antibody-based methods. Thus we think it is not justified to spend money and effort to validate results from a highly reliable method with results from a poorer method. We hoped the reader can see our validation is based on the completely independent multi-omics approach, as all the different data tell the same story. We consider the analysis of consistent molecular patterns much more relevant than any validation based on selected molecules.

Reviewer 3 Report

The authors analyzed mass spectrometric data of cerebrospinal fluid (CSF) from 8 (only 6 patients were used in analysis) recurrent medulloblastoma patients and 7 control patients. Based on these limited cases, the authors concluded that some molecules and biological pathways were upregulated in CSF of tumors compared to the control group, including hypoxia related genes and amino acids, lipolysis, and anti-inflammatory signaling. The major issues for this studies include 1) it is challenging to make a conclusion based on such a few samples without further validations; 2) the key genes related to the function pathways have to be validated in vitro and in vivo; 3) the identified tumor features are very common for cancer, but not unique for MB or any subtypes of MB; 4) molecular subtypes of MB are very important for patient outcome, tumor biology and therapy selection, the authors should define the subtypes of these tumors for analysis; 5) the quality of figures are very poor and all the images are not readable clearly. Other minor issues include 1) please use italic style for gene symbol; 2) MB_6 is lost in Figure 2A; 3) Line 135 “heat maps” is “heatmaps”; 3) Line 136 “PCA” doesn’t need explanation because it has been mentioned in Line 121;

Author Response

We would like to thank this reviewer for the helpful comments. Please find enclosed our point to point reply.

The major issues for this studies include 1) it is challenging to make a conclusion based on such a few samples without further validations;

  • This is a pilot study raising the question whether multi-omics might be a sensible strategy to investigate such complex disease. We are suggesting the answer was yes.We hoped the reader can see our validation is based on the completely independent multi-omics approach, as all the different data tell the same story. We consider the analysis of consistent molecular patterns much more relevant than any validation based on selected molecules.

2) the key genes related to the function pathways have to be validated in vitro and in vivo;

  • This represents a rather unhumble request. We think it was possible to make a single high-ranking paper out of each single regulatory gene when successfully validated in vitro and in vivo. Including such data into a single manuscript would result in a rather huge thing not yet seen in this world up until now.

3) the identified tumor features are very common for cancer, but not unique for MB or any subtypes of MB;

  • we have never claimed or stated otherwise. FSTL5 is described as a marker for MB, all the others as clearly indicated and cited. What was the problem here? It is well recognized ourdays that there hardly exists a single biomaker for a single disease, biomarkers are representative for specific pathomechanisms rather than specific diseases.

4) molecular subtypes of MB are very important for patient outcome, tumor biology and therapy selection, the authors should define the subtypes of these tumors for analysis;

  • We have done so (Table S4). As outlined in the Introduction, we were investigating the microenvironment rather than tumor cell properties. We think it is exciting to see such uniform deregulations in the tumor microenvironment irrespective of the genetic background of the disease. This may have great relevance for furture development of therapies.

5) the quality of figures are very poor and all the images are not readable clearly.

  • The low resolution is only a property of the files for review. Obviously we have high-resolution images and only high-resolution images will be published.

Other minor issues include 1) please use italic style for gene symbol;

  • We do not agree. We referring to proteins by using corresponding gene names (as they are short), this is commonly accomplished this way

2) MB_6 is lost in Figure 2A;

  • MB_6 was omitted as it was an outlier as indicated in the text

3) Line 135 “heat maps” is “heatmaps”;

  • corrected

3) Line 136 “PCA” doesn’t need explanation because it has been mentioned in Line 121;

  • You are right. However, we think this way increases readability

Round 2

Reviewer 1 Report

This reviewer considers the answers provided by the authors to be valid for the publication of their pilot study.

Reviewer 2 Report

I accept the manuscript in the present form.